# Th2 Suppression Through Antigen Liver Expression Using mRNA-LNP Technology

**DOI:** 10.3390/biomedicines13092297

**Published:** 2025-09-19

**Authors:** Kazunori Arai, Hanae Toyonaga, Lei Cheng, Hirotsugu Tanaka

**Affiliations:** 1Astellas Pharma, Inc., Tsukuba 305-8585, Japan; hirotsugu.tanaka@astellas.com; 2Astellas Pharma, Inc., Cambridge, MA 02141, USA; hanae.toyonaga@astellas.com; 3Astellas Pharma, Inc., Westborough, MA 01581, USA; lei.cheng@astellas.com

**Keywords:** mRNA-LNP, liver antigen expression, Th2 suppression

## Abstract

**Background:** Messenger RNA-lipid nanoparticle (mRNA-LNP) is a cutting-edge nucleic acid intracellular delivery technology. Although the clinical use of the mRNA vaccine is being actively developed, the use of mRNA-LNP technology in common diseases such as allergies is still being investigated. The purpose of this study is to test if immune response can be suppressed when an antigen is expressed in mice liver tissue with mRNA-LNP technology. **Methods:** We first designed mRNA which the ovalbumin (OVA) antigen expresses on the surface of the cells, and synthesized mRNA were encapsulated into LNP. This OVA-mRNA-LNP was evaluated with an OVA-sensitized mouse model. Splenocytes from OVA-sensitized mice were cultured with ex vivo OVA stimulation for Th2 cytokine production and Treg population analysis. Furthermore, OVA-mRNA-LNP was evaluated by both prophylactic and therapeutic administration in an OVA-induced mice airway inflammation model. **Results:** Th2 cytokines such as IL-4 and IL-5 were suppressed and the Treg population was increased in ex vivo OVA-stimulated splenocytes isolated from the OVA-mRNA-LNP administered group. Moreover, suppression of Th2 cytokines in Bronchoalveolar Lavage Fluid (BALF) from both the prophylactic and therapeutic OVA-mRNA-LNP administered cohort was observed (40–80% reduction in Th2 cytokines). **Conclusions:** The data suggests that mRNA-LNP technology, which is a safe, non-viral gene delivery system, can be an effective approach to suppress allergen-induced inflammation by expressing antigen in the liver tissue.

## 1. Introduction

mRNA-based gene delivery has become a promising platform for therapeutic drugs, showing significant potential in preventing and treating various diseases. The success of COVID-19 vaccines accelerated the development of mRNA technologies not only in infectious diseases but also in common diseases such as cancer, cardiovascular disease, and respiratory disease [1,2,3]. The one advantage of mRNA therapeutics is that this technology enables the druggability of intracellular, transmembrane, and secreted proteins without risk of genomic integration unlike viral gene delivery [4]. Several biotech companies or researchers are also actively developing mRNA delivery technology for the effective delivery of mRNA into cells or tissues. LNPs currently represent a nonviral mRNA delivery platform, consisting of four lipid components (ionizable lipid, polyethylene glycol (PEG)-lipid, phospholipid, and cholesterol) [5].

An allergy is an overreaction of our immune system to a specific substance called an allergen. In the past few decades, development of treatments for allergic diseases has increased worldwide. However, currently, there is no cure for allergies. One approach to cure allergies is preventing immune response caused by an antigen by expressing the antigen in the liver, leading to Treg population increase. Even though the biology is not completely understood, several papers show the important role of the liver microenvironment in the induction of liver tolerance [6,7,8]. Also, portal vein tolerance has been reported, characterized by the administration of antigens via the portal vein, which induces tolerance and accounts for the occurrence of tolerogenic responses in the liver [9]. Moreover, Chan et al. reported a potential treatment method for allergies by injecting OVA-coded adeno-associated virus (AAV) in OVA-sensitized mice. In this paper, suppression of the Th2-driven immune response and induction of Tregs were observed when delivering the OVA gene to the liver tissue [10]. However, due to its genome integration risk, antigen delivery by AAV vector is not suitable for common diseases considering the risk and benefit balance. Furthermore, mRNA-LNP allows for multiple administrations, enabling control over the expression levels [11,12,13].

In this paper, we demonstrate that OVA-mRNA-LNP can suppress Th2-driven immune response and induce the Treg population in OVA-sensitized mice. These results suggest that non-viral mRNA-LNP is an attractive technology for discovering drugs for allergy prevention and treatment.

## 2. Materials and Methods

### 2.1. Constructing Design for OVA mRNA Synthesis

The OVA coding sequence was fused with B7-1 domain including the transmembrane domain to generate membrane-bound OVA (Appendix A).

### 2.2. mRNA Synthesis

CleanCap^®^ Firefly luciferase (Fluc) mRNA (N1mΨ, RP-HPLC purified) and CleanCap^®^ OVA mRNA (N1mΨ, RP-HPLC purified) were synthesized at TriLink Biotechnologies (San Diego, CA, USA).

### 2.3. LNP Formulation

LNP formulation was conducted using NanoAssemblr^®^ Benchtop (Precision Nanosystems, Vancouver, BC, Canada). Briefly, ionizable lipid [10], the helper lipid DSPC (NOF America, White Plains, NY, USA, MC-8080), cholesterol (Nacalai USA, San Diego, CA, USA, NS460303), and PEG lipid DMG-PEG2000 (NOF America, GM-020) were dissolved in absolute ethanol at a molar ratio of 50:10:38.5:1.5. mRNA was prepared in a 6.25 mM acetate buffer (pH 5.0) at a concentration of 0.15 mg/mL. The lipid mixture and mRNA were mixed at an N/P ratio of 5.67 using a NanoAssemblr^®^ cartridge, with a total flow rate of 12 mL/min and a flow rate ratio of 3:1 (aqueous phase–organic phase). After formulation, the LNPs underwent buffer exchange with 1× DPBS (pH 7.4) and were concentrated using an Amicon^®^ Ultra-15 centrifugal filter unit (100 kDa NMWL, Millipore Sigma, Burlington, MA, USA), followed by filtration through a PVDF 0.22 μm filter (Millipore Sigma).

### 2.4. Ribogreen Assay

The mRNA encapsulation efficiency and concentration were assessed using the Quant-iT™ RiboGreen™ RNA Assay Kit (Thermo Fisher Scientific, Waltham, MA, USA, R11490). Briefly, LNPs were diluted with a 1× TE working solution, either in the absence or presence of 1% Triton X-100. mRNA standard curves were generated in the range of 0.3 to 5.0 μg/mL using the 1× TE working solution with 1% Triton X-100. The mixture was incubated at 37 °C for 20 min, followed by the addition of an equal volume of 1× Quant-iT™ RiboGreen™ RNA reagent to each sample. Fluorescence was measured using a SpectraMax^®^ device (Molecular Devices, San Jose, CA, USA) at an excitation wavelength of 485 nm and an emission wavelength of 525 nm. The mRNA encapsulation efficiency was calculated using the following method: Encapsulation Efficiency (%) = [(Fluorescence) Total mRNA − (Fluorescence) Outside mRNA]/(Fluorescence) Total mRNA × 100.

### 2.5. Cell Culture

The hepatocellular carcinoma cell line (Hepa1-6) was obtained from ATCC (Manassas, VA, USA). The cells were cultured in DMEM with 10% FBS and 1% penicillin–streptomycin and cells were grown under the conditions of 37 °C, 5% CO_2_.

### 2.6. Immunocytochemistry

Hepa1-6 cells were seeded in a 96-well plate at a density of 5 × 10^4^ cells per well and cells were treated with mRNA-LNP (0.1–10 µg/mL) or transfected with OVA-mRNA with Lipofectamine^®^ MessengerMax™ transfection reagent (Life Technologies, Carlsbad, CA, USA, LMRNA003). After one-day incubation, cells were fixed in 4% PFA for 10 min. After three washes, cells were treated with 0.1% Triton for permeabilization for 10 min followed by three washes and 1 h blocking with a blocking buffer. The cells were incubated with primary rabbit polyclonal antibody anti-OVA (Abcam, Waltham, MA, USA, ab18688) or control rabbit polyclonal antibody (Abcam, ab171870) for 1 h at room temperature. After three washes with PBS, the samples were incubated with Goat anti-rabbit IgG-IRDye 680RD-conjugated secondary antibody (Licor, Lincoln, NE, USA, 925-68071) for 30 min at room temperature. After three washes, images were recorded with Odyssey CLx (Licor).

### 2.7. Flow Cytometry

Flow cytometry was performed to analyze the expression of the OVA on the cell surface. Hepa1-6 cells were seeded in a 24-well plate at a density of 2 × 10^5^ cells per well and incubated for 5 h. Then, the cells were treated with mRNA-LNP (1 µg/mL) and incubated overnight. After incubation, the cells were trypsinized, washed, and prepared for flow cytometry analysis. The cells were incubated with primary rabbit polyclonal antibody anti-OVA (Abcam, ab18688) or control rabbit polyclonal antibody (Abcam, ab171870) for 1 h at room temperature and after three washes, the samples were incubated with Goat anti-rabbit IgG (H+L)-Alexa Fluor 488^®^-conjugated secondary antibody (Abcam, ab171870) for 1 h at room temperature. Expression of OVA with a FACSLyric™ flow cytometer (Becton–Dickinson, Franklin Lakes, NJ, USA) and FlowJo software (version 10.10.0, Becton-Dickinson) was used for analysis. The background signal was determined using matched isotype control.

### 2.8. Animal Study

The study was performed at Biomodels AAALAC accredited facility in Watertown, MA, USA. Approval for this study was obtained from Biomodels IACUC. During the study period, the animals were observed daily in order to reject animals that presented in poor condition. The animals were housed in HEPA-filtered, individually ventilated cages with bedding and provided with enviro-dri and a shepherd shack (enrichment). The animal samples were collected immediately following blood collection, after the animals were euthanized using an overdose of xylazine. The number of animals were determined by taking into account the experimental variability and the necessity to reduce the number of animals used.

### 2.9. OVA-Induced Inflammation Mouse Model

Female BALB/c nude mice were purchased from Charles River (Wilmington, MA, USA) for the development of the OVA-induced inflammation model at Biomodels. Briefly, on Day 0 and again on Day 7, the animals were sensitized with 20 µg OVA (Sigma-Aldrich, Saint Louis, MO, USA, A5503) plus 1.5 mg Alum (Aluminum hydroxide, Sigma-Aldrich, 239186) in 100 µL PBS via an intraperitoneal (IP) injection. The control group received no injections on Day 0 or Day 7. For the mechanism analysis study, 0.3 mg/kg mRNA-LNP was administered on Day −14, −12, −10, −8 via intravenous (IV) injection prior to OVA sensitization. On Day 28, the animals were euthanized, and spleens were collected for further analysis. For the OVA-induced airway inflammation model study, the animals were sensitized with 20 µg OVA plus 1.5 mg Alum in 100 µL PBS via IP injection followed by OVA antigen injection on Day 28, 29, and 30 with 20 µg OVA via an intranasal (IN) injection. A total of 50 µg/head OVA was injected via subcutaneous (SC) administration as SCIT (subcutaneous immunotherapy) control or 0.3 mg/kg mRNA-LNP was injected via IV either on Day −14, −12, −10, −8 for preventive treatment and on Day 14, 16, 18, 20 for therapeutic treatment. On Day 31, the animals were euthanized and blood/tissue/BALF samples were collected for further analysis.

### 2.10. Cytokine Measurement

Splenocytes were isolated by mincing the tissues into a single-cell suspension using a mesh filter. Red blood cells were lysed using ACK Lysing buffer for 1 min, followed by washing with RPMI 1640. The cells were spread in a 96 well plate at a density of 8 × 10^5^ cells per well and incubated for 96 h with 10 µg/mL OVA stimulation. The supernatant was collected after incubation for cytokine measurement. An ELISA kit (Mouse IL-4 Quantikine ELISA Kit, R&D, M400B and Mouse IL-5 Quantikine ELISA Kit, R&D Systems, Minneapolis, MN, USA, M5000) was used to measure IL-4 and IL-5 following their protocols. The IL-4 and IL-5 concentration in BALF was measured using a multiplex system analyzer (MAGPIX, EMD Millipore, Burlington, MA, USA).

### 2.11. FACS (Fluorescence-Activated Cell Sorting)

Isolated splenocytes were analyzed for Treg population and IL-10 expression. The cells were fixed and permeabilized with a Fixation/Permeabilization buffer (eBioscience, San Diego, CA, USA, 00-5523-00). Then, after washing with a 1× permeabilization buffer, the cells were incubated with antibodies (CD4-PerCP/Cy5: BD 550954, Foxp3-PE eBioscience, 12-5773-82, CD137-APC: eBioscince, 17-1371-82, IL-10-FITC: Thermo Fischer, A15376) for 1 h at 4 °C. Then, the cells were washed with 1× permeabilization buffer and measured with a FACSLyric™ flow cytometer (Becton–Dickinson). FlowJo software was used for analysis. The background signal was determined using matched isotype control.

### 2.12. Histopathology

Histopathologic evaluation included qualitative and semi-quantitative evaluation for pulmonary inflammation. Lung samples were fixed in 10% neutral buffered formalin. Samples were trimmed into four cross sections per sample/animal. Blocks were sectioned at approximately 5 µm and stained with Hematoxylin and Eosin (H&E) or Periodic acid-Schiff (PAS). Glass slides were evaluated by a board-certified veterinary pathologist using light microscopy. Histologic lesions were graded for severity (0 = absent; 1 = minimal; 2 = mild; 3 = moderate; 4 = marked; 5 = severe).

### 2.13. Statistics

Dunnett’s multiple comparisons test, Dunn’s multiple comparisons test, and an unpaired *t*-test were performed. All statistical analyses were performed using GraphPad Prism 8 software (version 10.0.3, La Jolla, CA, USA).

## 3. Results

### 3.1. mRNA-LNP Evaluation

mRNA was formulated into LNP by NanoAssemblr followed by buffer exchange with ultrafiltration. The encapsulation efficiency of mRNA-LNP used in this study was over 90%, measured by Ribogreen assay. A high expression of OVA was confirmed in mRNA-LNP-treated mouse liver cell line Hepa1-6 cells by ICC for both intercellular and cell surface expression and by FACS for cell surface expression (Figure 1).

### 3.2. Effect in OVA-Induced Inflammation Mouse Model

Th2 cytokines, Tregs, and IL-10 were analyzed from animals who were sensitized with OVA after we administered OVA-LNP or Fluc-LNP four times. Th2 cytokines were analyzed from cultured splenocytes with OVA stimulation. A slight decrease in IL-4 production and a significant decrease in IL-5 were observed in the OVA-LNP-injected group compared to the Fluc-LNP-injected group (Figure 2). For Tregs and IL-10-producing cells, non-cultured splenocytes were analyzed by FACS. A slight increase in IL-10-producing (CD4+, IL-10+) cells and a significant increase in the Tregs (CD4+, Foxp3+, CD137+) cell were observed in the OVA-LNP-injected group compared to the Fluc-LNP-injected group (Figure 3).

### 3.3. Effect in OVA-Induced Airway Inflammation Mouse Model

The design of the study is shown in Figure 4. The histopathological score and Th2 cytokines in BALF were analyzed to observe the effect of OVA-LNP. The SCIT administration showed a significant improvement in the inflammation score in the perivascular/peribronchiolar and interstitial area for both preventive and therapeutic administration. Also, for the inflammation score in the alveolar area, preventive SCIT administration and therapeutic SCIT administration showed a slight improvement. However, OVA-LNP administration did not show improvement in the inflammation score (Figure 5). Th2 cytokine concentrations in BALF were significantly decreased for IL-4 and slightly decreased for IL-5 in the preventive administration group and a slight decrease was observed in the therapeutically administered group. The Fluc-OVA administration group showed no suppression in Th2 cytokines. For the SCIT administered group, preventive administration showed a significant decrease in IL-4 and a slight decrease in IL-5 (Figure 6).

## 4. Discussion

The mRNA was synthesized with N1-methylpseudouridine modification to prevent unwanted inflammation as the replacement of uridine with N1-methylpseudouridine can reduce the immunogenicity of the resulting mRNA [14]. In addition to nucleoside modification, to eliminate double stranded RNA (dsRNA) which is a byproduct of mRNA synthesis by in vitro transcription, HPLC-purified grade mRNA was used in this study as dsRNA is also reported to mediate an immune response [15]. It is important to control the mRNA quality, especially when dosing repeatedly in animal studies to prevent off-target effects. Although further study is needed for the actual biodistribution of LNP, the liver is the primary tissue of LNP accumulation following intravenous administration [16], and LNP which was used in this study is known to be delivered to the liver [17].

Although the mechanism of Treg induction by antigen expression in the liver is not clear, it is reported that nonparenchymal cells such as dendritic cells, liver sinusoidal endothelial cells (LSECs), Kupffer cells, and hepatic stellate cells (HSCs) regulate immunosuppressive responses by IL-10 and TGF-b anti-inflammatory cytokine production [18,19,20,21,22]. And it is also known that IL-10 and TGF-β are importantly involved in the maintenance of immune suppression through Treg induction [23]. Several reports show that Tregs play an important role in Th2 suppression [24,25]. There was only a slight increase for IL-10 in CD4+ cells in the OVA-LNP-injected cohort in this study design, so the mechanism is not clear for the Treg induction in the ex vivo antigen stimulation experiment, but the inhibition of Th2 cytokine production was suppressed both in the OVA airway inflammation model and the OVA sensitizing model, in addition to the Treg induction effect observed in the OVA sensitizing model. Th2 activation with ex vivo antigen stimulation was suppressed in the splenocyte OVA-LNP-injected cohort, suggesting that expressing antigen in the liver induces the systemic Th2 suppressive effect. A detailed analysis of the mechanisms underlying Th2 suppression is necessary, utilizing techniques such as FACS and qPCR to assess antigen expression in various cell types, Treg profiles, and the levels of key mediators (e.g., TGF-β and IL-10) in the liver and spleen. One potential consideration is that Tregs may play a crucial role in Th2 suppression, indicating that Tregs could have been involved in suppressing Th2 responses in this study.

Several groups previously reported that the immune response can be suppressed and Tregs can be induced against a specific antigen when expressing antigen in the liver in mice [1,2]. Chan et al. reported liver-specific OVA expression by AAV2/8 vector virus suppressed airway hyper-responsiveness and OVA-specific IgE in bronchoalveolar lavage fluid (BALF) in OVA-sensitized mice [10].

The result of the OVA-induced airway inflammation model shows that Th2 suppression in BALF evaluating IL-4 and IL-5 cytokine production both in preventive and therapeutic administration is possible by delivering an antigen to the cell surface by mRNA-LNP technology. The absence of observed pathological improvement in the mRNA-LNP-treated cohort may be attributed to the potential innate immune activation induced by the LNPs themselves. In this study, we did not perform the profiling of early innate immune cytokines and complements, nor did we compare different administration routes to distinguish lipid-driven inflammation from antigen-specific tolerance. To mitigate the lipid-derived innate immune activation for clinical applications, it is essential to optimize the formulation [26].

It is reported that allergen-specific immunotherapy, such as SCIT, induces immunological changes, including the alleviation of allergen-specific Th2 immune responses, the induction of Th1 immune responses, the induction of Tregs, and the production of allergen-specific IgG4 antibodies [27,28]. There are also reports of the induction of regulatory B cells and the production of allergen-specific IgA antibodies, as well as the induction of IL-10-producing ILCs [29,30]. These complex effects of SCIT which are not limited to Th2 inhibition might be the reason that only the SCIT administration group demonstrated efficacy even in severe mouse models. Although the mouse model we selected is very severe this model is suitable to observe the clear Th2 effect. Although this study did not establish a cohort combining mRNA-LNP and SCIT, SCIT is currently being used for allergen-specific immunotherapy [31], and it may have a synergistic effect based on the underlying mechanisms. Additionally, dexamethasone may also serve as a valuable control when evaluating therapeutic efficacy to confirm clinical usefulness.

The histopathological study of goblet cell hyperplasia was also evaluated in addition to perivascular and alveolar changes (Appendix A). However, no significant score improvement was observed, even in the SCIT control group, despite histological changes in perivascular and alveolar structures in the SCIT group. Induction of goblet cell hyperplasia is known to be associated with Th2 immune responses [32], but the evaluation protocol (e.g., model severityand timepoint) may not be suitable for this assessment. When conducting efficacy evaluations with clinical significance, it is essential to optimize the evaluation models, including the timing of assessments, in future studies.

In this mouse model, antigens were administered twice, during the sensitization and the challenge, but it is possible that preventive administration before the sensitization reduces the immune response at the time of sensitization. This may explain why a stronger effect was observed in the preventive administration group compared to the therapeutic administration group.

To confirm not only the tissue distribution but also the specific cell types involved, biodistribution studies utilizing IHC (immunohistochemistry) or FACS are necessary for further investigation. Additionally, elucidating which cells in the liver are responsible for the mechanisms that induce Tregs is crucial for future research.

mRNA-LNP technology is a non-viral gene delivery system that offers advantages such as safety and high efficiency [4]. The success of the development of the COVID-19 vaccine accelerated the mRNA-LNP technology platform discovery and, currently, the drug discovery for multiple disease areas is actively ongoing globally. Although, for drug discovery, optimization of a delivery system by LNP screening and mRNA optimization (e.g., codon and UTR sequence) is necessary, the observations in this study show the potential of mRNA-LNP technology as a treatment for common diseases such as allergies.

## Figures and Tables

**Figure 1 biomedicines-13-02297-f001:**
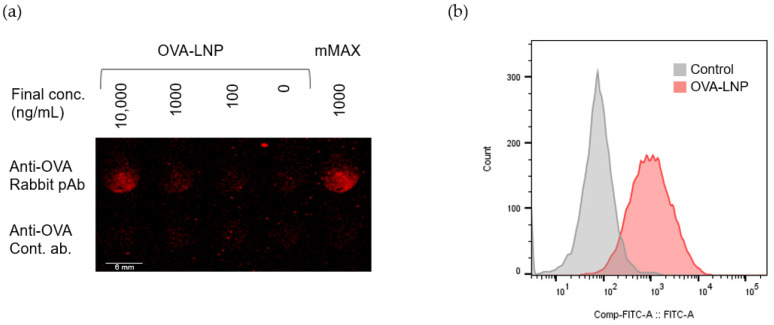
OVA expression in Hepa1-6 cells. OVA expression was evaluated by ICC (**a**) and FACS (**b**). OVA expression was analyzed after 24 h of OVA-LNP treatment or OVA-mRNA transfection.

**Figure 2 biomedicines-13-02297-f002:**
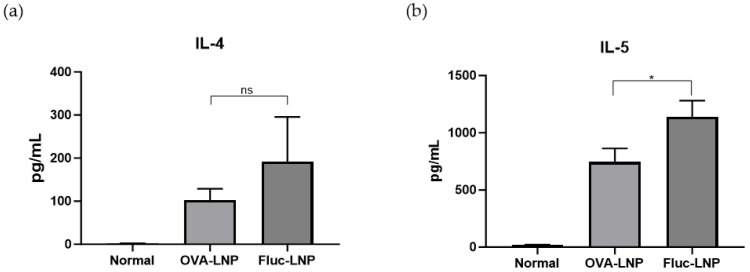
Cytokine production in splenocytes derived from OVA-induced inflammation mice. IL-4 (**a**) and IL-5 (**b**) concentrations were measured in a supernatant of OVA-stimulated splenocytes. Data is presented as mean ±SEM. *p*-value was determined using *t*-test; * = *p* < 0.05, ns = not significant.

**Figure 3 biomedicines-13-02297-f003:**
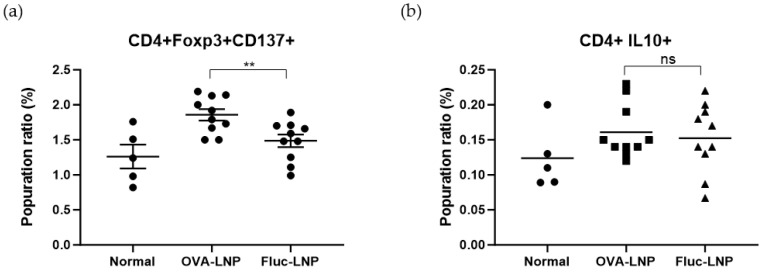
FACS analysis of splenocytes derived from OVA-induced inflammation mice. The cell population in the splenocytes was analyzed by FACS. Tregs (Foxp3+, CD137+) (**a**) and IL-10-producing cells (**b**) were analyzed in the CD4+ population. Data is presented as mean ±SEM. *p*-value was determined using *t*-test; ** = *p* < 0.01, ns = not significant.

**Figure 4 biomedicines-13-02297-f004:**
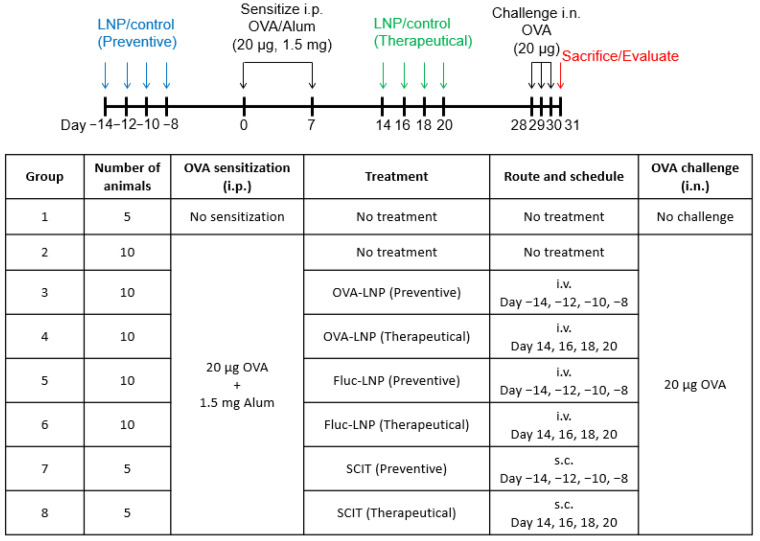
Study design and groups. The days are based on the initial sensitization (e.g., Day −14 means 14 days before initial sensitization). OVA-LNP: ovalbumin-lipid nanoparticles, OVA: ovalbumin, Alum: aluminum hydroxide, Fluc-LNP: Firefly luciferase-lipid nanoparticles, SCIT: subcutaneous immunotherapy, i.p.: intraperitoneal administration, i.v.: intravenous administration, s.c.: subcutaneous administration, i.n.: intranasal administration.

**Figure 5 biomedicines-13-02297-f005:**
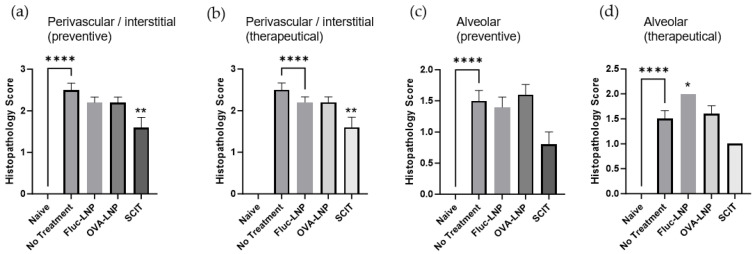
Mouse lung; mean histopathology scores. Inflammation scores are analyzed in the perivascular/peribronchiolar and interstitial (preventive administration: (**a**), and therapeutic administration: (**b**)), and the alveolar (preventive administration: (**c**) and therapeutic administration: (**d**)) area. Histologic lesions were graded for severity (0 = absent; 1 = minimal; 2 = mild; 3 = moderate; 4 = marked; 5 = severe). Group mean ± standard error of the mean (SEM). *p*-value was determined using the *t*-test for comparison between the no treatment group and the naïve group. For the other groups, in order to compare with the no treatment group, the *p*-value was determined using Dunn’s multiple comparisons test. * = *p* < 0.05, ** = *p* < 0.01, **** = *p* < 0.0001.

**Figure 6 biomedicines-13-02297-f006:**
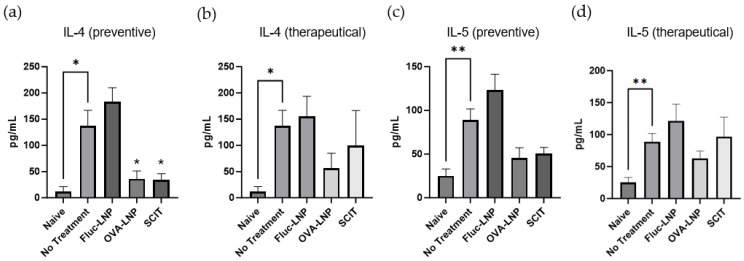
Cytokine concentration in BAL fluid. Inflammatory mediators were measured in BAL fluid. IL-4 (preventive administration: (**a**), and therapeutic administration: (**b**)) and IL-5 (preventive administration: (**c**) and therapeutic administration: (**d**)) data is presented as mean ± SEM. *n* = 5–10. *p*-value was determined using the *t*-test for comparison between the no treatment group and the naïve group. For the other groups in order to compare with the no treatment group, the *p*-value was determined using Dunnett’s multiple comparisons test. * = *p* < 0.05, ** = *p* < 0.01.

## Data Availability

Data can be obtained by request.

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
