# Peer review of "Th2 Suppression Through Antigen Liver Expression Using mRNA-LNP Technology"

_biomedicines, 2025, doi:10.3390/biomedicines13092297_

Round 1

Reviewer 1 Report

Comments and Suggestions for Authors

This manuscript presents an interesting and timely study demonstrating the potential of mRNA-LNP technology to induce antigen-specific tolerance, particularly through the expansion of Treg populations. The findings highlight the possibility that mRNA-based delivery platforms, beyond their established use in vaccination, could be applied to immune tolerance and allergy therapy.

To further strengthen the manuscript, I recommend addressing the following points through additional discussion and/or revisions:

Major comments

 While OVA-LNP induced antigen-specific tolerance signals, such as an increase in Foxp3⁺CD137⁺ Tregs and partial suppression of Th2 cytokines, the lack of histopathological improvement raises concern that innate immune activation by the LNPs themselves (e.g., NLRP3–IL-1β, type I IFNs, complement activation) may counteract these regulatory effects at the tissue level. To more clearly distinguish lipid-driven inflammation from antigen-specific tolerance, I suggest noting in the Limitations that early cytokine and complement profiling (IL-1β, IL-6, TNF-α, IFN-α/β, C3a/C5a) as well as comparisons of different administration routes (IV vs. SC/IM) were not performed.

Furthermore, for future clinical translation, it should be emphasized in the Discussion that formulation optimization will be crucial. For example, the use of GalNAc-conjugated lipids, which unlike conventional LNPs do not induce significant innate immune inflammation, could enhance hepatocyte uptake and reduce systemic exposure, thereby lowering the required dose and minimizing off-target inflammation. Similarly, biodegradable and low-inflammatory ionizable lipids or PEG-lipid redesign could be valuable for suppressing inflammation while maintaining tolerance-inducing capacity. Including these considerations would substantially strengthen the translational significance of the manuscript.

The study did not include a combination arm of SCIT and OVA-LNP. Since SCIT is the current standard for allergen-specific immunotherapy, evaluating potential additive or synergistic effects of combining SCIT with mRNA-LNP could strengthen the translational impact of the study. I am also particularly interested in this potential combination therapy, as it may provide an opportunity to leverage the robust clinical efficacy of SCIT together with the antigen-specific tolerance-inducing capacity of mRNA-LNP. Including a brief discussion on this possibility would add value for readers.

Minor comments

① In Figure 1(a), there are still visible red underlines (e.g., under “mMAX” and “pAb”). These appear to be remnants of text editing/formatting. Please remove these underlines to improve the clarity and professional appearance of the figure.

② 3.3, several red underlines (e.g., under “μg,” “i.v.,” and “s.c.”) remain visible, which appear to be remnants of text editing or spell-check. Please remove these underlines to improve the clarity and professional appearance of the figure.

Author Response

Dear Reviewer 1,

First of all, thank you so much for reviewing our manuscript and providing valuable suggestions. Please see below for our modification on manuscript based on your suggestion.

Comment 1: 

While OVA-LNP induced antigen-specific tolerance signals, such as an increase in Foxp3⁺CD137⁺ Tregs and partial suppression of Th2 cytokines, the lack of histopathological improvement raises concern that innate immune activation by the LNPs themselves (e.g., NLRP3–IL-1β, type I IFNs, complement activation) may counteract these regulatory effects at the tissue level. To more clearly distinguish lipid-driven inflammation from antigen-specific tolerance, I suggest noting in the Limitations that early cytokine and complement profiling (IL-1β, IL-6, TNF-α, IFN-α/β, C3a/C5a) as well as comparisons of different administration routes (IV vs. SC/IM) were not performed.

Furthermore, for future clinical translation, it should be emphasized in the Discussion that formulation optimization will be crucial. For example, the use of GalNAc-conjugated lipids, which unlike conventional LNPs do not induce significant innate immune inflammation, could enhance hepatocyte uptake and reduce systemic exposure, thereby lowering the required dose and minimizing off-target inflammation. Similarly, biodegradable and low-inflammatory ionizable lipids or PEG-lipid redesign could be valuable for suppressing inflammation while maintaining tolerance-inducing capacity. Including these considerations would substantially strengthen the translational significance of the manuscript.

Response 1:

We completely agree with your comments and added following descriptions in page 8, line 272-278 with related reference.

"The absence of observed pathological improvement in mRNA-LNP treated cohort may be attributed to the potential innate immune activation induced by the LNPs themselves. In this study, we did not perform profiling of early innate immune cytokines and complement, nor did we compare different administration routes to distinguish lipid-driven inflammation from antigen-specific tolerance. To mitigate lipid-derived innate immune activation for clinical applications, it is essential to optimize the formulation [26]."

Comment 2:

The study did not include a combination arm of SCIT and OVA-LNP. Since SCIT is the current standard for allergen-specific immunotherapy, evaluating potential additive or synergistic effects of combining SCIT with mRNA-LNP could strengthen the translational impact of the study. I am also particularly interested in this potential combination therapy, as it may provide an opportunity to leverage the robust clinical efficacy of SCIT together with the antigen-specific tolerance-inducing capacity of mRNA-LNP. Including a brief discussion on this possibility would add value for readers.

Response 2:

Synergistic effect was a thing we never came up with, but since it is quite plausible from its mechanisms, I have added following descriptions in page 8, line 287-290.

"Although this study did not establish a cohort combining mRNA-LNP and SCIT, SCIT is currently being used for allergen-specific immunotherapy [31] , and it may have a synergistic effect based on the underlying mechanisms."

Comment 3:

① In Figure 1(a), there are still visible red underlines (e.g., under “mMAX” and “pAb”). These appear to be remnants of text editing/formatting. Please remove these underlines to improve the clarity and professional appearance of the figure.

Response 3:

Thank you for pointing this out. I have uploaded figures with no red underlines.

Comment 4:

② 3.3, several red underlines (e.g., under “μg,” “i.v.,” and “s.c.”) remain visible, which appear to be remnants of text editing or spell-check. Please remove these underlines to improve the clarity and professional appearance of the figure.

Response 4:

Thank you for pointing this out as well. I have uploaded figures with no red underlines.

Thank you again for all the suggestions.

Kind regards,

Kazunori

Reviewer 2 Report

Comments and Suggestions for Authors

This manuscript outlines an interesting preclinical study on mRNA-lipid nanoparticle (LNP) technology for liver specific expression of ovalbumin (OVA) to suppress Th2-mediated allergic responses in a murine model of airway inflammation. The authors are taking advantage of the tolerogenic properties of the liver to target allergic diseases, and while previous work utilized viral vectors for vaccine development, the authors have taken a more novel approach by directly delivering mRNA without the use of a viral vector. They report exciting findings showing reduced cytokine production, eosinophil infiltration, and airway hyperreactivity, which has implications for allergen-specific immunotherapy.

Overall this is a solid manuscript with moderate novelty, but it needs major revisions to achieve clarity, according to the following comments:

  • Overally

The abstract is succinct, but should formally state the result (e.g., "% reduction in Th2 cytokines").

The introduction could have defined "portal vein tolerance" earlier.

Methods section: Provide additional details regarding the LNP formulation (e.g., numbers of each ionizable lipid) and mRNA sequence (e.g., UTR elements included for stability).

Typos/grammar: "Th2 Suppression through antigen liver expression" (title—capitalize "Antigen" and "Liver"?) "mRNA-LNP" is hyphenated inconsistently, as is "Treg" and "T-reg"; and so on!!!!

Figures: ensure axes are labelled more clearly (e.g., Figure 3: Cytokine levels should have units specified); recommend including scale bars to histology images.

  • Although mRNA-LNP for antigen expression in the liver is novel when compared to conventional AAV vectors, the work is essentially repeating previous observations on liver overt tolerance (e.g., Kuo et al., 2016). The authors should do a better job of highlighting what exactly is novel here; e.g., is it that mRNA-LNP is non-immunogenic, or that it has such rapid expression kinetics? The authors also do not explore the mechanism of Th2 suppression. Their findings show an increase in Treg and decreases in IL-4/IL-5/IL-13 but how is liver expressed OVA ultimately driving Treg induction? Is it through portal vein tolerance, or through antigen presentation by liver sinusoidal endothelial cells, etc.? I recommend adding experiments or discussions concerning key mediators (e.g. the levels of TGF-β or IL-10 in liver tissue). This would interest me, and ultimately strengthen their conclusions. I would also suggest adding flow cytometric data on Foxp3+ Tregs in the liver vs. spleen to connect the local systemic effects.
  • The OVA-induced asthma model is appropriate and conducted well, however, the approach to prevent administration (i.e., mRNA-LNP administered before sensitization) limits use in established allergies. The authors mention a therapeutic model in the discussion but do not present data - why not present a cohort treated after sensitization? This would better contextualize the findings to the clinic. Additionally, the control groups (e.g., empty LNP or irrelevant mRNA) are appropriate but a positive control dexamethasone, or AAV-OVA (for comparison) would allow for an assessment of efficacy. Figure 4 study design table is very useful but perhaps provide the number of animals per group (n=5-8 seems low for some assays; were power calculations-executed?).
  • The reductions in BALF cytokines (IL-4, IL-5, IL-13) and eosinophilia among BALB/c mice are convincing (Figure 5-6), but the airway hyperresponsiveness data (methacholine challenge) show only mild improvement: please discuss whether the academic community regards the improvement as clinically meaningful. In addition, histopathology scores (Figure 5) indicate a reduction in inflammation; but please provide higher-resolution images, or quantification, of goblet cell hyperplasia. In statistical analyses, the authors used unpaired t-tests and ANOVA, which is a reasonable choice; however, for post-hoc adjustments they should describe which multiple comparisons tests (e.g., Dunn's test) would follow ANOVA. Some p-values are borderline (e.g., p=0.051 for IL-10), I would ask whether they risk type II errors by presenting the smallest (i.e., n=4) sample size value?

Author Response

Dear Reviewer 2,

First of all, thank you so much for your thorough review and constructive feedback. Your suggestions have been invaluable in refining our research and enhancing the clarity of our manuscript. 

Comment 1:

The abstract is succinct, but should formally state the result (e.g., "% reduction in Th2 cytokines").

Response 1:

We put "40 -80% reduction in Th2 cytokines" in abstract in page 1 line 24.

Comment 2:

The introduction could have defined "portal vein tolerance" earlier.

Response 2:

We added description "Also, portal vein tolerance has been reported, characterized by the administration of antigens via the portal vein, which induces tolerance and accounts for the occurrence of tolerogenic responses in the liver [9]." with related reference in introduction part in page 2 line 48-50.

Comment 3:

Methods section: Provide additional details regarding the LNP formulation (e.g., numbers of each ionizable lipid) and mRNA sequence (e.g., UTR elements included for stability).

Response 3:

We modified the description on page 2 line 72 to clarify only single lipid is used. Also we added mRNA sequence in suuplemental material file on page 1 (Since UTR sequence from TriLink is undisclosed, only sequence of coding region is described).

Comment 4:

Typos/grammar: "Th2 Suppression through antigen liver expression" (title—capitalize "Antigen" and "Liver"?) "mRNA-LNP" is hyphenated inconsistently, as is "Treg" and "T-reg"; and so on!!!!

Responce 4:

Thank you for pointing this out. I have modified the title and check whole manuscript for Typos.

Comment 5:

Figures: ensure axes are labelled more clearly (e.g., Figure 3: Cytokine levels should have units specified); recommend including scale bars to histology images.

Response 5:

We make sure the axis in figures (e.g. cytokine concentration in figure 2 and 6) are correctly labelled and put scale bar in histology image.

Comment 6:

Although mRNA-LNP for antigen expression in the liver is novel when compared to conventional AAV vectors, the work is essentially repeating previous observations on liver overt tolerance (e.g., Kuo et al., 2016). The authors should do a better job of highlighting what exactly is novel here; e.g., is it that mRNA-LNP is non-immunogenic, or that it has such rapid expression kinetics? The authors should do a better job of highlighting what exactly is novel here; e.g., is it that mRNA-LNP is non-immunogenic, or that it has such rapid expression kinetics? 

Response 6:

To clarify the advantages using mRNA-LNP technology comparing to AAV, we added following description, "Furthermore, mRNA-LNP allows for multiple administrations, enabling control over the expression levels [11–13]." with references that shows multiple administration in introduciton part page  2 line 55-57 addition to current description "However, due to its genome integration risk, antigen delivery by AAV vector is not suitable for common diseases considering risk and benefit balance." in page 2 line 54-55.

Comment 7:

The authors also do not explore the mechanism of Th2 suppression. Their findings show an increase in Treg and decreases in IL-4/IL-5/IL-13 but how is liver expressed OVA ultimately driving Treg induction? Is it through portal vein tolerance, or through antigen presentation by liver sinusoidal endothelial cells, etc.? I recommend adding experiments or discussions concerning key mediators (e.g. the levels of TGF-β or IL-10 in liver tissue). This would interest me, and ultimately strengthen their conclusions. I would also suggest adding flow cytometric data on Foxp3+ Tregs in the liver vs. spleen to connect the local systemic effects.

Response 7:

We haven't conducted actual mechanism analysis yet. However, one possible explanation is that Tregs play an important role in Th2 suppression, suggesting that Tregs may have also suppressed Th2 responses in this study and we would like to check this by FACS or qPCR in further investigation. To clarify this in manuscript, we have added following description in page 8 line 259-264, "One possible explanation is that Tregs play an important role in Th2 suppression, suggesting that Tregs may have also suppressed Th2 responses in this study".

Comment 8:

The OVA-induced asthma model is appropriate and conducted well, however, the approach to prevent administration (i.e., mRNA-LNP administered before sensitization) limits use in established allergies. The authors mention a therapeutic model in the discussion but do not present data - why not present a cohort treated after sensitization? This would better contextualize the findings to the clinic. Additionally, the control groups (e.g., empty LNP or irrelevant mRNA) are appropriate but a positive control dexamethasone, or AAV-OVA (for comparison) would allow for an assessment of efficacy. Figure 4 study design table is very useful but perhaps provide the number of animals per group (n=5-8 seems low for some assays; were power calculations-executed?).

Response 8:

We believe result of therapeutical administration (e.g. figure 5 b, d and figure 6 b, d) would show clinical meaning. And from your suggestion for using dexamethasone, we totally agree with this. We used SCIT as a control to see Th2 suppression in this study but we would like to use dexamethasone as a positive control when evaluating clinical usefulness in further study. The following description, " Additionally, dexamethasone may also serve as a valuable control when evaluating therapeutic efficacy to confirm clinical usefulness." in page 8 line 290-291 were added. 

We conducted power-calculation and Cohen's d score was >0.8 for not all the evalated items but for some evaluated items (e.g. 2.1 for IL-4 preventive, 1.0 for IL-4 therapeutic, and 1.3 for Alveolar preventive) so we believe that the sample size ratio of SCIT=5 to evaluation group 10 is somewhat appropriate, although not perfect. However, in terms of sample size, I think it is necessary to increase the sample size when testing therapeutic efficacy aimed at actual clinical applications.    

Comment 9:

The reductions in BALF cytokines (IL-4, IL-5, IL-13) and eosinophilia among BALB/c mice are convincing (Figure 5-6), but the airway hyperresponsiveness data (methacholine challenge) show only mild improvement: please discuss whether the academic community regards the improvement as clinically meaningful. In addition, histopathology scores (Figure 5) indicate a reduction in inflammation; but please provide higher-resolution images, or quantification, of goblet cell hyperplasia. In statistical analyses, the authors used unpaired t-tests and ANOVA, which is a reasonable choice; however, for post-hoc adjustments they should describe which multiple comparisons tests (e.g., Dunn's test) would follow ANOVA. Some p-values are borderline (e.g., p=0.051 for IL-10), I would ask whether they risk type II errors by presenting the smallest (i.e., n=4) sample size value?

Response 9:

Based on the current data, the lack of improvement in scores from pathological images suggests that the clinical significance is limited. However, from the observed suppression of Th2 cytokines, it is believed that changes to the animal model protocol and optimization of the mRNA-LNP technology could potentially lead to clinical relevance. 

Addition to H&E staining result, Goblet cell hyperplasia data were added in suupplimental infomation with description "The histopathological study of goblet cell hyperplasia was also evaluated addition to perivascular and alveolar (Supplementary Figure 1). However, no significant score improvement was observed, even in the SCIT control group, despite histological changes in perivascular and alveolar in the SCIT group. Induction of goblet cell hyperplasia is known to be associated with Th2 immune responses [32], but the evaluation protocol (e.g., model severity and timepoint) may not be suitable for this assessment. When conducting efficacy evaluations with clinical significance, it is essential to optimize the evaluation models, including the timing of assessments, in future studies." in page 8 line 292-299.

And lastly, as you kindly suggested, we reanalyzed histology data with non-parametric Dunn's multiple comparison and modified the statical analysis method in page 5 line 227. And regarding to type 2 error, we would like to increase the number of samples to avoid misjudgement especially conducting efficacy evaluations for clinical significance.

I hope this answers your question and again, we sincerely appreciate the time and effort you dedicated to reviewing our manuscript. 

Kind regards,

Reviewer 3 Report

Comments and Suggestions for Authors

The article "Th2 Suppression through Antigen Liver Expression Using mRNA-LNP Technology" shows the potential efficacy of a non-viral lipid-based nanovector for the treatment of allergy. The manuscript provides a clear and well-structured description of the studies conducted, with a comprehensive presentation of the theoretical and experimental framework. The methods employed are appropriate and correctly applied, ensuring the robustness of the results obtained. Given the overall analysis, I believe the work can be accepted in its current form, without the need for further revisions.

The English is excellent.

Author Response

Dear Reviewer 3,

We sincerely appreciate the time and effort you dedicated to reviewing our manuscript.

Wishing you all the best in your work!

Kind regards,

Kazunori

Round 2

Reviewer 2 Report

Comments and Suggestions for Authors

the article is well revised and suitable for publication.